# Occupational Difficulties of Disaster-Affected Local Government Employees in the Long-Term Recovery Phase after the Fukushima Nuclear Accident: A Cross-Sectional Study Using Modeling Analysis

**DOI:** 10.3390/ijerph19073979

**Published:** 2022-03-27

**Authors:** Yuya Kashiwazaki, Hitomi Matsunaga, Makiko Orita, Yasuyuki Taira, Keiko Oishi, Noboru Takamura

**Affiliations:** Department of Global Health, Medicine and Welfare, Atomic Bomb Disease Institute, Nagasaki University Graduate School of Biomedical Sciences, Nagasaki 8528523, Japan; hmatsu@nagasaki-u.ac.jp (H.M.); orita@nagasaki-u.ac.jp (M.O.); y-taira@nagasaki-u.ac.jp (Y.T.); tak011004@gmail.com (K.O.); takamura@nagasaki-u.ac.jp (N.T.)

**Keywords:** Fukushima, nuclear disaster, local government employee, job satisfaction, radiation anxiety, sense of coherence

## Abstract

Local government officials play a central role in post-disaster community reconstruction. However, few studies have reported on the actual difficulties during a complex disaster involving a nuclear accident. A self-rated questionnaire survey was administered to a total of 583 public employees in four municipalities around the Fukushima Daiichi Nuclear Power Station. The relationship between universal occupational factors and radiation disaster-specific factors on job satisfaction and intention to leave the job due to radiation anxiety was evaluated using structural equation modeling. The results showed that interpersonal problems (β = −0.246) and service years (β = −0.127) were related to job satisfaction, whereas radiation-specific factors were not related to job satisfaction, and only to the intention to leave work due to radiation anxiety. A sense of coherence was associated with job satisfaction (β = 0.373) and intention to leave work due to radiation anxiety (β = −0.182), and it served as a moderator of the universal occupational factors and the radiation disaster-specific factors. Therefore, it is suggested that outcomes could be improved through increased stress coping capacity by providing support for relationships and radiation risk communication to public employees during the disaster recovery period.

## 1. Introduction

When a disaster occurs, public employees are engaged in disaster response as part of their duties, which has been shown to place a heavy burden on their bodies and minds [1,2,3]. Especially in the acute phase after the nuclear disaster at the Fukushima Daiichi Nuclear Power Station (FDNPS) in 2011, the functions of the local government itself were moved to other districts as evacuation orders were issued, and the staff themselves had to work in an unfamiliar environment, separated from their families, even though they were victims of the disaster. Just as the previous Chernobyl nuclear disaster pointed out the psychosocial problems of the affected residents [4], the FDNPS accident showed similar problems [5,6,7], and the local government employees were no exception [8]. In a survey of public employees in municipalities near the FDNPS, it was reported that these public employees were at high risk of depression [9], and support for public employees who play a fundamental role in regional reconstruction was recognized as an urgent issue. In addition, one of the unique tasks related to the nuclear disaster was that public employees were required not only to perform their general reconstruction efforts as duties, but also to respond to the radiation concerns of residents, namely, to engage in radiation risk communication. Naturally, even the employees of the local governments near the nuclear power plants did not have any knowledge about the health effects of radiation, and they had to provide support while experiencing the same anxiety as the residents. After the disaster, experts provided support in terms of physical and mental health, as well as basic knowledge about radiation and alleviating related anxiety [10,11]. In supporting community reconstruction, the importance of collaboration among experts, government, and local residents while respecting the local climate and culture was proposed, rather than unilateral involvement by experts (co-expertise process) [12]. In particular, public employees play a core role in disaster recovery, and it is worth considering how to support them.

Therefore, this study focused on sense of coherence (SOC) as a factor contributing to the mental health of municipal employees. This concept was proposed based on the fact that some people suffer from poor health and others maintain good health when placed in stressful situations. SOC is defined as “a global orientation that expresses the extent to which one has a pervasive and enduring, though dynamic, feeling of confidence” based on the salutogenesis of how this health and well-being can be restored, maintained, and enhanced [13]. Moreover, SOC is composed of three sub-factors: (1) the stimuli deriving from one’s internal and external environments in the course of living are structured, predictable, and explicable (comprehensibility); (2) adequate resources are available to one to meet the demands posed by these stimuli (manageability); and (3) these demands are challenges, and are worthy of investment and engagement (meaningfulness) [13]. Individuals with high SOC are reported to have a good capacity for coping with stress [14]. Therefore, SOC can be expected to function as a factor for alleviating the occupational stressors of public employees during the disaster recovery phase.

As a result of the FDNPS accident following the Great East Japan Earthquake, approximately 16,000 residents of Fukushima Prefecture were evacuated, and 10 years later, approximately 34,000 residents are still living as evacuees inside and outside Fukushima [15]. In addition, the return rate of residents varies by municipality, and in many cases, some of the functions of the public offices are located in the evacuated destination areas. The purpose of this study was to identify the problems faced by these employees of the disaster-affected municipalities at different stages of recovery and to provide knowledge that will contribute to their support. More concretely, the relationships between universal occupational and radiation-specific factors on job satisfaction and intention to leave work due to radiation anxiety, and how SOC can mitigate these outcomes for public employees in four municipalities around the FDNPS (Kawauchi Village, Tomioka Town, Okuma Town, and Futaba Town; Figure 1) were examined.

The hypotheses of this study were as follows: (1) suffering due to relationship problems, perceived workload, and distress from radiation counseling decrease job satisfaction; (2) radiation-specific factors such as perception of the risk of genetic effects of radiation, deterioration of health due to radiation, and distress from radiation counseling increase intention to leave the job due to radiation anxiety; and (3) SOC increases job satisfaction and decreases intention to leave the job due to radiation anxiety.

## 2. Materials and Methods

### 2.1. Study Participants

A self-administered survey of local government employees in four towns and villages adjacent to the FDNPS, namely Kawauchi Village, Tomioka Town, Okuma Town, and Futaba Town, was conducted. The total number of survey participants was 583: 68 in Kawauchi Village, 178 in Tomioka Town, 172 in Okuma Town, and 165 in Futaba Town. The survey period was from 1 July to 30 July 2021, and a total of 514 (88.2%) participants responded to the survey. Of these, 24 participants were excluded from the analysis because of missing data, and 490 (84.0%) were included for the analysis.

### 2.2. Measures

The objective variables of this study were job satisfaction and intention to leave the job due to radiation anxiety. Job satisfaction is considered to be one of the important variables of subjective well-being [16], and the participants answered the question “Are you satisfied with your current workplace?” on a scale of “1. satisfied” to “4. not satisfied”. Regarding intention to leave the job due to radiation anxiety, the participants were asked to answer on a binary scale of “yes” or “no” to the question, “Have you ever wanted to leave the public office where you currently work because you were worried about the health effects of radiation?”.

They were also asked about their sex and age as basic demographic data, their service years, whether they thought that their workload was heavy compared to other municipal employees, and whether they had ever had problems with interpersonal relationships in the workplace. In addition, as radiation-related items, they were asked if they had ever been troubled by radiation consultations from residents, if they had ever felt that their health had deteriorated due to exposure to radiation from the accident, and if they thought that there were genetic effects due to exposure from the accident. The responses to the above first two items used a binary scale of “yes” or “no”, whereas the responses regarding radiation risk perception used a 4-point Likert scale ranging from “1. likely” to “4. unlikely”.

For SOC, the Japanese version of the Sense of Coherence scale was used [13]. This scale is a 13-item, 7-point Likert scale with a 3-factor structure of “comprehensibility”, “manageability”, and “meaningfulness”, and its reliability and validity have been confirmed [14,17]. These factors are strongly related to each other, and it has been suggested that they should be taken as a single concept [17,18]. However, because the validity of each factor has been demonstrated in previous studies [17], and to ensure ease of interpretation and practicality, they were treated as variables with a three-factor structure in this study. Examples of SOC questions are as follows: “1. Do you have the feeling that you don’t really care about what goes on around you? (meaningfulness)”; “2. Has it happened in the past that you were surprised by the behavior of people whom you thought you knew well? (comprehensibility)”; and “3. Has it happened that people whom you counted on disappointed you? (manageability)”.

### 2.3. Statistical Analysis

The basic characteristics of the overall sample and of the participants of each town and village, the frequency and percentage of the two objective variables, and the arithmetic mean, standard deviation, and Cronbach’s α coefficient of the SOC scale were confirmed. The associations between these variables and each town and village were also confirmed using the chi-squared (χ^2^) test and one-way analysis of variance. Next, the χ^2^ test was used to check for explanatory variables related to the two objective variables, and Student’s *t*-test was used for the SOC scale. Note that, only for the χ^2^ test, the 4-point Likert scale questions were combined at the midpoint into 2 points for calculation. Finally, the analysis was performed using structural equation modeling (SEM). The model was built based on the hypotheses, and SOC was arranged as the moderator for each explanatory variable for the outcomes (Figure 2). The three sub-factors of SOC were used as observed variables, and SOC was used as a latent variable. Only the factor of service years was used as a binomial scale, with at least 10 years and less than 10 years determined to be meaningful groups, and the other variables were analyzed according to the measured scales. The objective variable of intention to leave the job due to radiation anxiety was a binary “yes” or “no” variable, which was treated as an ordinal scale for convenience of analysis, and the estimation method used was diagonally weighted least squares. This estimation method uses the full weight matrix to compute robust standard errors and then adjusts the mean and variance to obtain a test statistic [19,20]. Model fit was evaluated with the χ^2^ statistic, the comparative fit index (CFI), the standardized root mean residual (SRMR), and the root mean square error of approximation (RMSEA). A model is typically accepted as an adequate fit when CFI > 0.90, and SRMR and RMSEA < 0.08 [21]. All statistical procedures were performed in R ver. 4.1.1 (R Foundation for Statistical Computing, Vienna, Austria) and R studio ver. 2021.09.0 (Integrated Development for R. PBC, Boston, MA, USA) software, and the SEM analysis package in R, lavaan 0.9–9, was used with a 0.05 significance level. The error variables in the SEM model figure have been omitted.

## 3. Results

The results of descriptive statistics for the basic characteristics, the two objective variables, and the moderating variable are shown in Table 1. There were no differences in these variables between towns and villages. Cronbach’s alpha coefficient of the SOC scale was 0.86, confirming that it was sufficiently reliable.

Next, the association of each variable with the objective variable is shown in Table 2. There were no associations of the sex and age groups with the outcomes. For job satisfaction, there were significant associations with service years, perception of workload, interpersonal problems, being troubled due to radiation consultations, intention to leave work due to radiation anxiety, and SOC. For intention to leave work due to radiation anxiety, there were significant associations with service years, job satisfaction, interpersonal problems, health deterioration due to radiation exposure, perception of the risk of genetic effects of radiation, and SOC.

Based on the results of univariate analysis and the hypothesis, the relationship between each variable was verified using SEM (Figure 3). The goodness of fit of the model was acceptable, with χ^2^ = 47.098, CFI = 0.914, RMSEA = 0.053, and SRMR = 0.042.

The contribution of each factor to SOC was 0.842 for comprehensibility, 0.895 for manageability, and 0.689 for meaningfulness. Being troubled by radiation consultations was not associated with job satisfaction, and each of the three radiation-specific factors, i.e., troubled by radiation consultations (β = 0.156), health deterioration due to radiation exposure (β = 0.218), and perception of risk of genetic effects of radiation (β = 0.245), was positively associated with intention to leave the job due to radiation anxiety. Universal occupational variables, i.e., length of service (β = −0.127) and interpersonal problems (β = −0.246), had a negative effect on job satisfaction. For SOC, radiation risk perception (β = −0.114) and, especially, troubled relationships (β = −0.518) showed strong negative associations, and SOC increased job satisfaction and mitigated intention to leave the job due to radiation anxiety. In addition, the perception of workload was related to job satisfaction on univariate analysis, but not in the SEM. Regarding the significant total effect of these explanatory variables, the effect of interpersonal problems on work satisfaction was −0.439 (−0.246: direct effect + [−0.518 × 0.373]: indirect effect), and that of service years was −0.148. In the same way, the effect of health deteriorating due to radiation on intention to leave the job due to radiation anxiety was 0.220, and that of perception of risk of genetic effects of radiation was 0.245.

## 4. Discussion

The purpose of this study was to identify the occupational difficulties of local government employees around the FDNPS during the long-term recovery period and to examine support measures. The four municipalities are in various phases of recovery, ranging from those where residents started returning to their homes one year after the accident to those where all residents are still living as evacuees, but there was no difference in job satisfaction and intention to leave the job due to radiation anxiety among those groups. The average SOC of the overall participants was 55.6, which is approximately the same or slightly lower than the previously reported SOC average of 59 in Japan [22]. Although the SOC of public employees was expected to be relatively high due to their stable employment, this result may be due to their difficulty in feeling confident that they will be able to cope with the various demands of the radiation disaster or that they will always have available the resources to do so. This is also an environment in which the prospects for reconstruction are difficult to predict.

For job satisfaction, it was found that longer tenure (more than 10 years of service) was associated with lower job satisfaction, with a rate of nearly 40%. This is because the employees who have been with the municipality for more than 10 years experienced the disaster while they were working there, and they are the group that actually suffered the difficulties associated with the evacuation order, such as having to live away from their families and commuting long distances, and the effects of these environmental changes are continuing. However, exposure to grievances from residents is a major risk factor for the mental health of public employees and should be taken into account, regardless of seniority. In addition, experiencing difficulties in radiation consultations and perception of high work volume were not significantly related to job satisfaction in the SEM. This indicates that the quality of human relationships in carrying out work is more important than distress about work volume or having special tasks. There is a theory that public service motivation defines the attitude that is characteristic of public servants [23,24]. This is a concept that expresses the wish to “help public and society”, and this altruism and the disposition to serve public interest are also assumed to be behind the fact that workload was not related to job satisfaction. However, it has been suggested that this disposition may also be related to the ease of stress accumulation, and it would be useful to include factors such as the characteristics of a public servant in future research.

Approximately 30% of the participants had the perception that radiation is associated with genetic risks. Although the percentage is considerably lower than that of the general residents in the region [25,26], it should be noted that 30% of the employees still had concerns. Nine percent of the overall study cohort and approximately 17% of the employees with 10 or more years of service had thought about leaving the job due to radiation anxiety. In fact, it is believed that some of them have resigned [27]. Furthermore, all three radiation-specific variables were associated with intention of leaving the job due to radiation anxiety. Feeling anxious about radiation itself is a natural human reaction, but practical problems such as leaving a job or unidentified complaints should be prevented. Although there is a limit to the effectiveness of communicating scientific knowledge about radiation risk perception, a certain effect has been shown [28,29], and radiation risk communication to stakeholders such as local government officials near a nuclear power plant is important.

SOC functioned as an alleviating factor for the two objective variables. In addition, having relationship problems was a strong predictor of SOC. SOC is considered to be an indicator of relatively stable intentionality rather than a state [13], and its association with distress in relationships may represent characteristic factors such as passivity in work communications and difficulty in building relationships. Several studies have reported that interventions can improve SOC [30,31]. Therefore, in addition to adjusting the hard aspects such as the working environment, SOC could be improved through the development of communication skills. A typical example is assertiveness training [32], and the skills for psychological recovery also include a section on human relations [33], which may be useful in terms of general mental health measures during disaster recovery. Furthermore, although SOC has been shown to have a positive effect on various health-related factors [34,35], it has also been associated with the perception that radiation has a risk of genetic effects, which are more specific factors. Previous work has shown ways to influence radiation risk perception by approaching individual health perspectives, such as general health anxiety [36]. In addition, the results of this study suggest that approaching radiation risk perception itself may also enhance stress acceptance and, in turn, alleviate the difficulties experienced in the workplace. This may be because radiation risk communication increases the sense of comprehensibility and manageability, which may alleviate anxiety about the unknowns of radiation [37]. Although the factor loadings were lower for meaningfulness than for the other factors, it would be important for public employees to be involved so that they can reaffirm the significance and importance of their work in regional revitalization as well.

There are some limitations of this study. First, because this was a cross-sectional study, the path directions in the SEM are hypothetical and do not indicate causality. Therefore, future studies using a longitudinal design are desirable to better clarify causal relationships. In addition, the actual effects of support need to be clarified through interventional studies. Second, since the survey was conducted 10 years after the disaster, it cannot be applied to the acute phase.

Despite the aforementioned limitations, the fact that more than 80% of the participants responded to the survey supports its validity in clarifying the actual situation of disaster-affected municipal employees, and this study provides meaningful information for examining how to support disaster-stricken municipal employees during the long-term recovery period. Few studies have focused on public officials, especially after a nuclear disaster, and further research is encouraged.

## 5. Conclusions

Public officials have among the most stressful occupations in the post-disaster recovery process. The relationships of universal occupational and radiation-specific factors with job satisfaction and the experience of leaving a job due to radiation anxiety among employees of the affected municipalities 10 years after the Fukushima disaster were examined. Universal occupational factors were associated with job satisfaction, and radiation-specific factors were associated with intention to leave the job due to radiation anxiety. SOC was found to have a buffering effect on these outcomes. Therefore, support for interpersonal communication skills in the workplace to improve stress coping capacity is important for the occupational well-being of municipal employees during the long-term recovery phase of a complex disaster. In addition, there is a need to provide knowledge about radiation and more proactive risk communication about radiation anxiety, including from the perspective of nuclear emergency preparedness.

## Figures and Tables

**Figure 1 ijerph-19-03979-f001:**
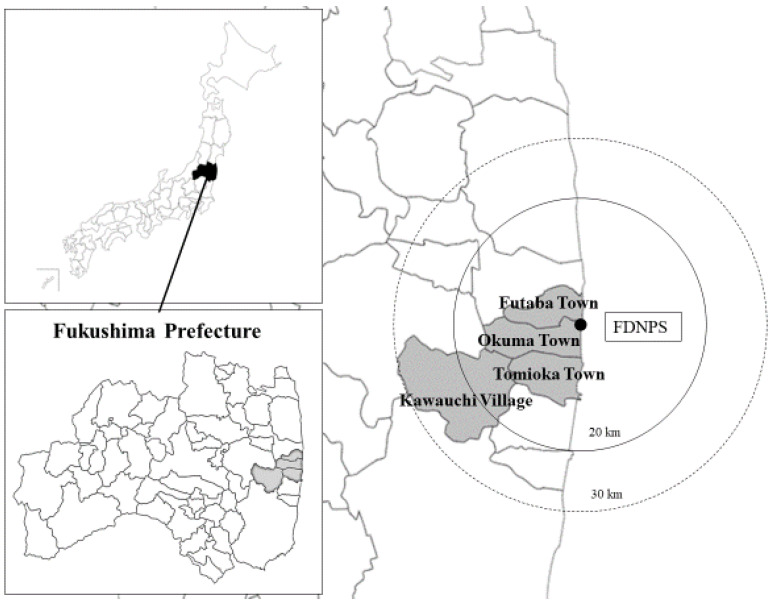
Location of the four municipalities in relation to the FDNPS.

**Figure 2 ijerph-19-03979-f002:**
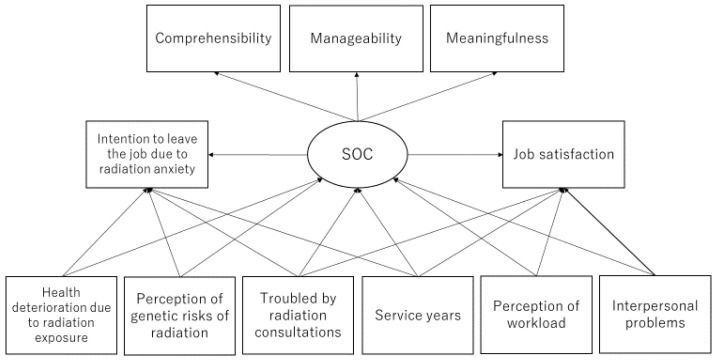
A hypothesis model of occupational difficulties of public employees in the Fukushima disaster area.

**Figure 3 ijerph-19-03979-f003:**
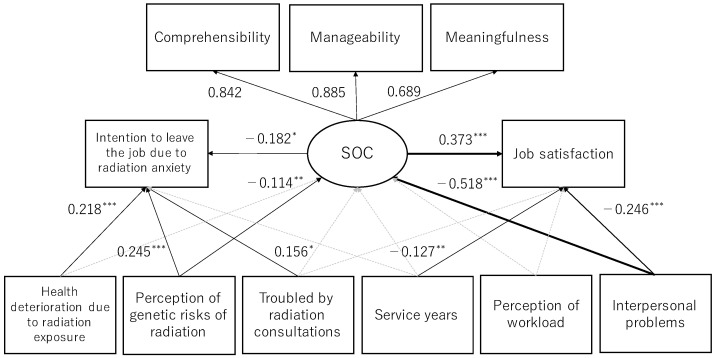
Results by structural equation modeling. All coefficients are standardized estimates, * *p* < 0.05, ** *p* < 0.01, *** *p* < 0.001. The circle is the latent variable, and the squares are observed variables. Paths are shown only if the coefficients are significant.

**Table 1 ijerph-19-03979-t001:** Descriptive analysis of demographic variables, objective variables, and the moderating variable.

	Overall		Kawauchi		Tomioka		Okuma		Futaba		*p*
	*n*	%	*n*	%	*n*	%	*n*	%	*n*	%	
Sex											
Male	331	67.6	34	75.6	104	66.7	108	66.7	85	66.9	0.69
Female	159	32.4	11	24.4	52	33.3	54	33.3	42	33.1	
Age											
<40 years old	195	39.8	15	33.3	75	48.1	57	35.2	48	37.4	0.08
≥40 years old	295	60.2	30	66.7	81	51.9	105	64.8	79	62.2	
Job satisfaction											
Satisfied	353	72	37	82.2	111	71.2	120	74.1	85	66.9	0.22
Not satisfied	137	28	8	17.8	45	28.8	42	25.9	42	33.1	
Intention to leave the job due to radiation anxiety											
Yes	44	9	5	11.1	18	11.5	13	8	8	6.3	0.42
No	446	91	40	88.9	138	88.5	149	91	119	93.7	
SOC ^†^											
mean/SD	55.6	13.2	60.6	13.9	53.6	13	55.9	11.4	55.9	14.8	0.68

^†^ = one-way analysis of variance, SD = standard deviation.

**Table 2 ijerph-19-03979-t002:** Relationships of each observed variable with the two objective variables.

	Overall (*n* = 490)		Job Satisfaction				Intention to Leave Work Due to Radiation Anxiety			
			Satisfied (*n* = 353)		Not Satisfied (*n* = 137)		Yes (*n* = 44)		No (*n* = 446)	
	*n*	%	*n*	%	*n*	%	*n*	%	*n*	%
Sex										
Male	331	67.6	237	71.6	94	28.4	27	8.2	304	91.8
Female	159	32.4	116	73	43	27	17	10.7	142	89.3
Age										
<40 years old	195	39.8	141	72.3	54	27.7	20	10.3	175	89.7
≥40 years old	295	60.2	212	71.9	83	28.1	24	8.1	271	91.9
Service years										
<10 years	305	62.2	241	79	64	21 ***	13	4.3	292	95.7 ***
≥10 years	185	37.8	112	60.5	73	39.5	31	16.8	154	83.2
Job satisfaction										
Satisfied	353	72	-	-	-	-	17	4.8	336	95.2 ***
Not satisfied	137	28	-	-	-	-	27	19.7	110	80.3
Perception of workload										
Much	296	60.4	203	68.6	93	31.4 *	33	11.1	263	88.9
Not much	194	39.6	150	77.3	44	22.7	11	5.7	183	94.3
Interpersonal problems										
Yes	281	57.3	165	58.7	116	41.3 ***	39	13.9	242	86.1 ***
No	209	42.7	188	90	21	10	5	2.4	204	97.6
Troubled by radiation consultations										
Yes	134	27.3	87	64.9	47	35.1 *	25	18.7	109	81.3 ***
No	356	72.7	266	74.7	90	25.3	19	5.3	337	94.7
Intention to leave the job due to radiation anxiety										
Yes	44	9	17	38.6	27	61.4 ***	-	-	-	-
No	446	91	336	75.3	110	24.7	-	-	-	-
Health deterioration due to radiation exposure										
Yes	29	5.9	20	69	9	31	14	48.3	15	51.7 ***
No	461	94.1	333	72.2	128	27.8	30	6.5	431	93.5
Perception of risk of genetic effects of radiation										
Yes	148	30.2	99	66.9	49	33.1	28	18.9	120	81.1 ***
No	342	69.8	254	74.3	88	25.7	16	4.7	326	95.3
SOC ^†^	mean	SD	mean	SD	mean	SD	mean	SD	mean	SD
	55.6	13.2	59	11.8	47	12.8 ***	46.7	13.6	56.5	12.8 ***

* *p* < 0.05, *** *p* < 0.001, ^†^ = Student’s *t*-test, SD = standard deviation.

## Data Availability

All data are available from the corresponding author on reasonable request.

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
