# Peer review of "Occupational Difficulties of Disaster-Affected Local Government Employees in the Long-Term Recovery Phase after the Fukushima Nuclear Accident: A Cross-Sectional Study Using Modeling Analysis"

_ijerph, 2022, doi:10.3390/ijerph19073979_

Round 1
Reviewer 1 Report
Well written manuscript. All sections are written appropriately. Approach is sound, methodology is sound. Conclusions adequately explain results. A few comments and questions are provided in the attached file.

Author Response
Response to Reviewer 1 Comments
Well written manuscript. All sections are written appropriately. Approach is sound, methodology is sound. Conclusions adequately explain results. A few comments and questions are provided in the attached file.
- Given that the study focuses on SOC, it may be useful to the reader to define or give examples of the three sub-factors: "comprehensibly," manageability," and "meaningfulness" (even though the reference is provided)
Response 1
We appreciate your comment and hope that the revised version addresses any concerns you may have. Explanations regarding Antonovsky’s SOC sub-factor definitions are included on P2L61-65.
“Moreover, SOC is composed of three sub-factors: 1) the stimuli deriving from one’s internal and external environments in the course of living are structured, predictable, and explicable (comprehensibility); 2) the resources are available to one to meet the demands posed by these stimuli (manageability); and 3) these demands are challenges, worthy of investment and engagement (meaningfulness) [13].”
- Was education considered as a demographic variable? If not, why? Education is often linked to understanding of radiation.
Response 2
Because this survey targeted a specific occupation, public service, it was expected that the majority of respondents would be high school or university graduates (relatively highly educated). Therefore, differences in results by education were not considered significant and were not included in the survey instrument.
- It may be useful to include a sample of the questions provided in the survey.
Response 3
We added a scale item example on P4L125-129.
“Examples of SOC questions are as follows: “1. Do you have the feeling that you don’t really care about what goes on around you? (meaningfulness)”; “2. Has it happened in the past that you were surprised by the behavior of people whom you thought you knew well? (comprehensibility)”; and “3. Has it happened that people whom you counted on disappointed you? (manageability)”.”
- Explain how the "validity" of each factor is ensured. How?
Response 4
We have added, "in previous studies" and reference [17] on P4L123. Furthermore, regarding the validity of using a three-factor structure, the results of the confirmatory factor analysis in the Structural Equation Modeling in this study confirmed certain factor loadings for each factor, although the value for meaningfulness was small. Therefore, "The contribution of each factor to SOC was 0.842 for comprehensibility, 0.895 for manageability, and 0.689 for meaningfulness." has been added to the Results section (P6L183-184).
- Statistical approach is appropriate
Response 5
Thank you for your confirmation.
Kind regards.
Reviewer 2 Report
Thank you for the opportunity to review this article. After an interesting read, there are some recommendations for improvement:
a) The article idea should be validated using a greater number of the newest scientific literature (I would suggest having more than half of your used references published after 2017). This would help to prepare the literature review part of your article and at the same time validate your theoretical model presented in figure 2.
b) I would suggest equating your used scales: you have used different scales to measure different research objects. Statistically, it is important to use the same scales to be able correctly to compare the data or use them for the same statistical calculations.
c) Conclusion part is too general and repeats the research results. It could be improved by determining specific results and their impact on science.
The article could be published after minor revision.
Author Response
Response to Reviewer 2 Comments
a) The article idea should be validated using a greater number of the newest scientific literature (I would suggest having more than half of your used references published after 2017). This would help to prepare the literature review part of your article and at the same time validate your theoretical model presented in figure 2.
Response1
We appreciate the time and effort you have dedicated to providing constructive comments on ways to improve our paper. We have added some of the latest literature that supports the argument of this study (References 6, 7, 8, 24, and 29.) We have also added to the discussion the paucity of research in this field (P9L283-284).
“Few findings have focused on public officials, especially after a nuclear disaster, and further research is encouraged.”
[References]
6.Takebayashi, Y.; Lyamzina, Y.; Suzuki, Y.; Murakami, M. Risk Perception and Anxiety Regarding Radiation after the 2011 Fukushima Nuclear Power Plant Accident: A Systematic Qualitative Review. Int. J. Environ. Res. Public Health 2017, 14, 1306, doi:10.3390/ijerph14111306.
7.Oe, M.; Takebayashi, Y.; Sato, H.; Maeda, M. Mental Health Consequences of the Three Mile Island, Chernobyl, and Fukushima Nuclear Disasters: A Scoping Review. Int. J. Environ. Res. Public Heal.2021, Vol. 18, Page 7478, 18, 7478, doi:10.3390/IJERPH18147478.
8.World Health Organization A framework for mental health and psychosocial support in radiological and nuclear emergencies.; Genova, 2020; Licence: CC BY-NC-SA 3.0 IGO
24.Wang, T.M.; van Witteloostuijn, A.; Heine, F. A Moral Theory of Public Service Motivation. Front. Psychol. 2020, 11, doi:10.3389/FPSYG.2020.517763.
29.Kashiwazaki, Y.; Takebayashi, Y.; Murakami, M. The relationship between geographical region and perceptions of radiation risk after the Fukushima accident: The mediational role of knowledge. Radioprotection 2022, 57, 17–25, doi:10.1051/RADIOPRO/2021027.
b) I would suggest equating your used scales: you have used different scales to measure different research objects. Statistically, it is important to use the same scales to be able correctly to compare the data or use them for the same statistical calculations.
Response 2
Thank you for your valuable feedback. As you pointed out, the fact that the measurements varied from one scale to another, especially that one of the response variables was measured in binary form, necessitated the use of a specific statistical method. We will try to standardize the measurements for future research.
c) Conclusion part is too general and repeats the research results. It could be improved by determining specific results and their impact on science.
Response 3
Thank you very much. As you noted, the conclusion was a reiteration of the results, and we have revised it to make the recommendations for support easier to understand (P9L286, L292-296).
“Public officials have among the most stressful occupations in the post-disaster recovery process.”; “Therefore, support for interpersonal communication skills in the workplace to improve stress coping capacity is important for the occupational well-being of municipal employees during the long-term recovery phase of a complex disaster. In addition, there is a need to provide knowledge about radiation and more proactive risk communication about radiation anxiety, including from the perspective of nuclear emergency preparedness.”
Kind regards.
Reviewer 3 Report
Figure 1 not at all clear as presented. The whole Fukushima area not clearly delineated as given.
Author Response
Response to Reviewer 3 Comments
- Figure 1 not at all clear as presented. The whole Fukushima area not clearly delineated as given.
Response 1
Thank you for your careful assessment and consideration of the figure presented in our paper. We have revised the figure to make it easier to understand the location of Fukushima Prefecture and the municipalities surveyed.
Kind regards